# Primary Processing and Storage Affect the Dominant Microbiota of Fresh and Chill-Stored Sea Bass Products

**DOI:** 10.3390/foods10030671

**Published:** 2021-03-22

**Authors:** Faidra Syropoulou, Foteini F. Parlapani, Stefanos Kakasis, George-John E. Nychas, Ioannis S. Boziaris

**Affiliations:** 1Laboratory of Marketing and Technology of Aquatic Products and Foods, Department of Ichthyology and Aquatic Environment, School of Agricultural Sciences, University of Thessaly, Fytokou Street, 38446 Volos, Greece; faisyropou@uth.gr (F.S.); fwparlap@uth.gr (F.F.P.); stkakasi@uth.gr (S.K.); 2Laboratory of Microbiology and Biotechnology of Foods, Department of Food Science and Human Nutrition, School of Food and Nutritional Sciences, Agricultural University of Athens, Iera Odos 75, 11855 Athens, Greece; gjn@aua.gr

**Keywords:** fish, seafood, spoilage, microbiota, primary processing, HRM, 16S rRNA gene sequencing

## Abstract

The cultivable microbiota isolated from three sea bass products (whole, gutted, and filleted fish from the same batch) during chilled storage and the effect of primary processing on microbial communities in gutted and filleted fish were studied. Microbiological and sensory changes were also monitored. A total of 200 colonies were collected from TSA plates at the beginning and the end of fish shelf-life, differentiated by High Resolution Sequencing (HRM) and identified by sequencing analysis of the V3–V4 region of the 16S rRNA gene. *Pseudomonas* spp. followed by potential pathogenic bacteria were initially found, while *Pseudomonas*
*gessardii* followed by other *Pseudomonas* or *Shewanella* species dominated at the end of fish shelf-life. *P. gessardii* was the most dominant phylotype in the whole sea bass, *P. gessardii* and *S. baltica* in gutted fish, while *P. gessardii* and *P. fluorescens* were the most dominant bacteria in sea bass fillets. To conclude, primary processing and storage affect microbial communities of gutted and filleted fish compared to the whole fish. HRM analysis can easily differentiate bacteria isolated from fish products and reveal the contamination due to handling and/or processing, and so help stakeholders to immediately tackle problems related with microbial quality or safety of fish.

## 1. Introduction

Aquaculture with its low carbon footprint is the most environmentally friendly source of animal protein for human nutrition, while traditional livestock such as beef, cattle, pig and poultry have been recognized as significant contributors to climate change due to the emissions of CO_2_ -eq at gigatonnes per annum [1,2]. Moreover, it has been characterized as the key to eradicate hunger and malnutrition worldwide even in the context of global population increase [3]. For Hellenes, aquaculture is a basic pillar of the national economy and one of the greatest weapons against recession. Greece is a significant world producer of farmed fish with production over 110 K tonnes annually, of which European sea bass (*Dicentrarchus labrax*) represents over 40% of production [4].

The impact of Hellenic fish production is high for the global food supply chain since approximately 80% of the total production is exported to the EU (e.g., Italy, Spain and France) and the rest of the world (e.g., USA, Canada and third countries). Fish such as European sea bass can be exported as whole, gutted, or filleted at low storage temperatures or in ice. However, tonnes of fish from aquaculture production are lost every year due to microbial spoilage and safety issues in pre- and post-farm gate, threatening food security and economy. During farming, harvesting, handling, primary processing and distribution, various microorganisms from the environment or workers might enter pre- or post-harvest handling/process and contaminate fish [5]. These microorganisms might be serious microbial spoilers of fish or seafood-borne pathogens. For gutted and filleted fish, contamination with such bacteria is more possible than the whole fish due to the use of tools and equipment in handling and processing.

In storage, a consortium of bacteria of the initial microbiota, the so-called Specific Spoilage Organisms (SSOs), can dominate over the rest of the microorganisms and produce metabolites responsible for the deterioration of sensory attributes and so the rejection of the product [6]. Meanwhile, the composition of the SSOs consortium can differ among products e.g., whole, gutted and filleted fish, due to a series of factors such as the composition of the initial microbiota (including the level and type of contamination), type of product, storage conditions, and microbial interactions [7]. Such differences can lead to different shelf-life of the products, even when they are stored under the same storage conditions, since different bacterial genera, species or strains can present different growth rates or metabolism.

To tackle problems related with microbial spoilage and safety in aquaculture and the fish processing industry, researchers attempt to find intelligent tools or toolkits for the rapid detection of microorganisms. High Resolution Melting (HRM) analysis has been used for identification of bacterial species or molecular typing [8,9,10,11]. In HRM, the amplicons can give similar melting curve profiles (fluorescence—temperature, >91% similarity) for the same bacterial species/strain [12]. Bacteria can be easily differentiated based on the shape of HRM normalized melting curves even when they present the same melting peak temperature [10,12]. Therefore, isolates can be grouped into the same bacterial species/strain and then identification is feasible by the sequencing of one or more representatives of each bacterial group, reducing the cost of sequencing [10,12]. This method has been proposed as a rapid method for differentiation of bacterial phylotypes, even those of a high degree of homology (including SSOs and hygiene indicators), in finfish and shellfish during storage [10,12].

The aim of this study was (a) to determine the cultivable microbiota isolated from three sea bass products (whole, gutted, and filleted fish from the same batch) during chilled storage, and (b) to study the effect of primary processing on microbial communities, including SSOs and hygiene indicators, in gutted and filleted European sea bass compared to the whole fish. In this context, HRM analysis was used for the differentiation of the cultivable microbiota, while the bacterial identification was performed by sequencing analysis of the V3–V4 region of the 16S rRNA gene.

## 2. Materials and Methods

### 2.1. Provision and Storage of Sea Bass

Three different European sea bass products (whole, gutted and filleted) were provided from a Greek aquaculture company in November 2019. The products were packaged in insulated boxes with melted ice and transferred to the laboratory of Marketing and Technology of Aquatic Products and Foods (University of Thessaly, Volos, Greece). The boxes of whole and gutted sea bass were stored at 0 °C for 18 days, while the ice was replaced every two days. The boxes of fillets were stored at 2 °C for 12 days. Fish were stored under the particular temperatures to simulate the commercial storage and distribution conditions.

### 2.2. Rejection Time

The sensory attributes (e.g., skin appearance, and odor of flesh) for the products were evaluated according to the Multilingual Guide to EU Freshness Grades for Fishery Products [13] by five trained panelists. Each panelist provided an independent evaluation. The rating of each sensory attribute was scored using a 5 to 1 scale, with 5 being the highest quality and 1 the lowest. A score of 3 was considered the score for minimum acceptability and the time point that average score was below 3 (which means that at least one out of the five panelists scored with 2) was considered the rejection time point, while a score of 1 was attributed to a totally spoiled sample. The aim of the evaluation was to determine the rejection time point of the three products. The evaluation was carried out every 3 days for the whole and gutted fish, and every 2 days for fish fillets.

### 2.3. Microbiological Analysis

In order to prepare the first decimal dilution, ten grams (10 g) of sea bass tissue were placed aseptically into a stomacher bag with 90 mL MRD (Maximum Recovery Diluent, 0.1% *w*/*v* peptone, 0.85% *w*/*v* NaCl) and homogenized for 3 min using a Stomacher (Bug Mixer, Interscience, London, UK). Enumeration of the following microorganisms was carried out with the spread plating method, using 0.1 mL of the serial dilutions in MRD: (1) aerobic plate count (APC) on TSA (Tryptone Soy Agar) incubated for 48–72 h at 25 °C and (2) *Pseudomonas* spp., on cetrimidefucidin-cephaloridine agar (CFC), incubated for 48 h at 25 °C. The enumeration of the following microorganisms was carried out with the pour plating method, using 1 mL of the serial dilution in MRD: (1) H_2_S-producing bacteria on Iron Agar (IA) by counting only black colonies, after incubation at 25 °C for 72 h, (2) Lactic Acid Bacteria on De Man, Rogosa, Sharpe agar (MRS) after incubation at 25 °C for 72 h and (3) Enterobacteriaceae on Violet Red Bile Glucose agar (VRBGA), incubated at 37 °C for 24 h. The analysis was performed every 3 days for the whole and gutted fish, and every 2 days for fish fillets. The results were expressed as mean log cfu g^−1^ ± standard deviation of 3 replicates (3 whole, gutted, or filleted fish per time point, 3 fish × 7 sampling points = 21 fish per product). Microbiological media were obtained from LAB M (Lancashire, UK) and Iron Agar was prepared according to Gram et al. [14].

### 2.4. Identification of Bacteria Isolated from the Sea Bass Products

#### 2.4.1. Isolation of Colonies and DNA Extraction

A percentage of 30–50% of the colonies grown on TSA plates (each one containing ~30–300 colonies) at the beginning and at the end of shelf life for each product (56, 72 and 72 colonies in total for the whole, gutted, and filleted fish, respectively), was randomly obtained for the molecular analysis. Each isolate was sub-cultured in TSA and incubated at 25 °C for 24–48 h.

Before the DNA extraction an overnight culture at 37 °C, of each isolate, in TSB was prepared. A volume of 200 μL of the culture was used for the DNA extraction with the Nucleospin Microbial DNA kit (Macherey-Nagel GmbH & Co. KG, Düren, Germany). The DNA samples were stored at −20 °C until the 16S rRNA gene analysis by HRM.

#### 2.4.2. HRM Analysis

The pair of universal primers S-DBact-0341-b-S-17 (5′-CCTACGGGNGGCWGCAG-3′) and S-D-Bact-0785-a-A-21 (5′-GACTACHVGGGTATCTAATCC-3′) according to Klindworth et al. [15] was used for the amplification of the V3–V4 region of the 16S rRNA gene on a PCR max Eco48 Real-time qPCR system (Cole-Parmer Ltd., Stone, UK). The reaction mixture (final volume of 20 μL) consisted of KAPA HRM FAST Master Mix 1X (Kapa Biosystems Pty Ltd., Cape Town, South Africa), 1.5 mM MgCl_2_, 0.2 μM of each primer and 20 ng genomic DNA. The PCR conditions were (for 40 cycles) as follows: a pre-PCR step (polymerase activation) at 95 °C for 3 min, denaturation at 95 °C for 5 s, annealing at 57 °C for 30 s and elongation at 72 °C for 40 s. After PCR amplification, HRM was performed with a pre-melt step at 92 °C for 15 s followed by a melt step from 75 to 92 °C. The experiments were performed in triplicate.

#### 2.4.3. Sequencing

The 50% of the isolates from each of the bacterial groups that presented both similar melting peak temperature and normalized melting curve shape in HRM, were sequenced. Sanger sequencing was performed using an ABI Prism 3730 XL capillary sequencer by CeMIA SA (Larissa, Greece). For each sample, forward and reverse reads were assembled. BLAST function was used for the detection of the closest relatives [16]. Sequence data were aligned using the Clustal Omega algorithm from Geneious Prime V2020.1.1 [17]. A cut-off level of 98% sequence similarity to GenBank entries was used to define the closest relatives of the sequences. Sequences of dominant phylotypes found in this study are presented as Appendix A.

### 2.5. Statistical Analysis

Differences of means in microbial counts were statistically tested. STATISTICA 6.0 was used for Analysis of Variance followed by Tukey’s significant difference test. A probability level of *p* ≤ 0.05 was considered statistically significant.

## 3. Results

### 3.1. Shelf-Life of Sea Bass Products

Fish freshness was excellent at the beginning of shelf-life for all three fish products (Figure 1). Skin was bright, shiny and iridescent, gills were bright red, eyes had convex lens and translucent cornea, and odor was fresh/seaweedy, in whole and gutted fish (for the latter the peritoneum was glossy, brilliant, and difficult to tear from flesh). For fillets, the appearance was translucent, glossy, presented a natural color and odor was marine and fresh. Sensory attributes diminished gradually with time.

At the time point of minimum acceptability (sensory score equal to 3), skin was dull with some bleaching, eyes had slightly concave and opaque cornea, gills were brown and odor was slightly rancid for whole and gutted fish (for the latter the peritoneum was gritty and easy to tear from flesh). Sea bass fillets represented an opaque and dull appearance and stale odor. Then, fish was degraded to unfit and rejected (sensory score less than 3). As determined by sensory assessment, the shelf-life of the whole, gutted, and filleted sea bass was 15, 15 and 10 days, respectively. The grade (5, 4, 3, 2, 1) of general appearance and odor for each product related with days of storage are shown in Figure 1.

### 3.2. Microbial Populations

At the beginning of fish shelf-life, APC was found at similar population levels (4.96 ± 0.35 and 5.07 ± 0.19 log cfu/g, respectively) for the whole and filleted fish (*p* > 0.05), but at lower population levels for the gutted fish (3.77 ± 0.07 log cfu/g; *p* ≤ 0.05). APC increased during the storage, reaching the level of 7.46 ± 0.33, 8.24 ± 0.76, and 8.19 ± 0.22 log cfu/g at the time of minimum acceptability level (day 15, 15 and 10; *p* ≤ 0.05 between whole and other fish products, *p* > 0.05 for gutted and filleted fish) and 7.90 ± 0.30, 8.88 ± 0.62, and 8.86 ± 0.15 log cfu/g at the rejection time point (day 18, 18 and 12) for the whole, gutted and filleted sea bass, respectively (Figure 2).

The populations counted on CFC, IA, MRS, and VRBGA were initially at the levels of 4.40 ± 0.12, 4.19 ± 0.28, 2.06 ± 0.08 and 2.33 ± 0.10 log cfu/g for whole fish, 3.41 ± 0.04, 2.41 ± 0.04, 1.85 ± 0.21 and 1.72 ± 0.34 log cfu/g for gutted fish, and 5.02 ± 0.22, 4.16 ± 0.04, 3.61 ± 0.43 and 1.95 ± 1.35 log cfu/g for fish fillets (Figure 2), thus showing that the populations on CFC and IA were much higher compared to those counted on MRS, and VRBGA (*p* ≤ 0.05), at the beginning of fish shelf-life. Additionally, gutted sea bass presented lower populations for all microorganisms tested compared to the whole and filleted fish (*p* ≤ 0.05). At the minimum acceptability level, the population on CFC, IA, MRS, and VRBGA were reached at the levels of 7.54 ± 0.13, 6.87 ± 0.12, 5.98 ± 0.28 and 6.33 ± 0.10 log cfu/g for whole fish, 8.28 ± 0.47, 7.73 ± 0.74, 4.57 ± 0.81 and 5.01 ± 0.44 log cfu/g for gutted fish, and 8.13 ± 0.08, 7.42 ± 0.24, 6.01 ± 1.01 and 5.82 ± 1.16 log cfu/g for fish fillet, presenting the domination of *Pseudomonas* sp. followed by H_2_S-producing bacteria for all three products (*p* ≤ 0.05). These were also the most dominant microorganisms at the end of fish shelf-life, reaching the levels of 7.79 ± 0.08, and 7.37 ± 0.23 log cfu/g, 8.87 ± 0.54, and 7.57 ± 0.81 log cfu/g and 8.76 ± 0.13, and 8.14 ± 0.07 log cfu/g for *Pseudomonas* sp. and H_2_S-producing bacteria in three products, respectively (Figure 2).

### 3.3. Bacterial Communities of Fresh and Chill-Stored Sea Bass Products

Two hundred (200) isolates presented 19 distinct HRM curve profiles (data not shown) and classified into 19 groups, based on both the melting temperature peaks and shape of normalized melting curves of the amplicons using HRM analysis (Table 1). Using the 16S rRNA gene sequencing analysis, the groups were found to consist of three different phylotypes of *Pseudomonas* sp. (SBS-FS3, SBS-FS4 and SBS-FS5), two phylotypes of *Shewanella putrefaciens* (SBS-FS7 and SBS-FS8), and phylotypes of *Pseudomonas putida*, *Pseudomonas gessardii*, *Pseudomonas fluorescens*, *Pseudomonas fragi*, *Aeromonas hydrophila*, *Stenotrophomonas maltophilia*, *Shewanella baltica*, *Bacillus paralicheniformis*, *Klebsiella oxytoca*, *Serratia fonticola*, *Serratia grimessi*, *Enterobacter ludwigii*, *Rouxiella* sp. and *Staphylococcus warneri* (Table 1). Of these phylotypes, some were found to be common between whole and processed sea bass, while others were present in whole or gutted and/or filleted fish, at the beginning and/or at the end of shelf-life (Figure 3).

Initially (day 0), *Pseudomonas* sp. (phylotype SBS-FS3) followed by *S. maltophilia*, *P. putida*, *A. hydrophila*, *K. oxytoca*, *B. paralicheniformis* and *S. fonticola* were the most dominant phylotypes for the whole sea bass (Figure 3). *P. putida* followed by *A. hydrophila*, *S. warneri*, *Pseudomonas* sp. (phylotype SBS-FS3), *P. gessardii*, *P. fluorescens*, *S. grimessi* and *E. ludwigii* dominated in fresh gutted fish, while *Pseudomonas* sp. (phylotype SBS-FS4) and *Rouxiella* sp. (in equal abundances) followed by *P. putida*, *P. fluorescens*, *A. hydrophila*, *Pseudomonas* sp. (phylotype SBS-FS3), *S. grimessi*, *S. warneri* and *E. ludwigii* dominated in fresh sea bass fillets (Figure 3).

At the end of fish shelf-life (day 15 for the whole and gutted fish and day 10 for fish fillets), *P. gessardii* at abundance of 55.6% followed by *Pseudomonas* sp. (phylotype SBS-FS3), *P. putida*, *S. baltica* and *S. putrefaciens* (phylotype SBS-FS8) at lower abundances, were the most dominant bacteria grown on TSA for the whole sea bass (Figure 3). *P. gessardii* and *S. baltica* (37.5% and 29.2% respectively) were found at higher abundances compared to the rest microbiota for the gutted fish, while *P. gessardii* and *P. fluorescens* (25.0% and 20.0% respectively) followed by *S. baltica*, *Pseudomonas* sp. (phylotype SBS-FS5), *Pseudomonas* sp. (phylotype SBS-FS4) and other species of *Pseudomonas* and *Shewanella* were the most dominant bacteria in sea bass fillets (Figure 3).

## 4. Discussion

Microorganisms such as *Pseudomonas*, *Bacillus* and *Aeromonas* can be usually found at the beginning of fish shelf-life [6], including the European sea bass [12]. In the present study, phylotypes belonging to *Pseudomonas* sp., *P. putida*, *P. gessardii*, *P. fluorescens*, *B. paralicheniformis* and *A. hydrophila* were initially found in the whole, gutted and/or filleted sea bass. Such bacteria usually originate from the environment e.g., seawaters, freshwaters or sediment, in which fish live [5]. Of these, *P. fluorescens* and/or *P. putida* have been also found to constitute the most predominant bacteria of gilthead sea bream [18,19,20] or other fish [21,22]. Based on HRM melting curve profiles, *Pseudomonas* sp. (phylotype SBS-FS3) was a common phylotype in all three sea bass products, while *Pseudomonas* sp. (phylotype SBS-FS4) was only found in fish fillets. *Pseudomonas* sp. strain SeaQual_P_B791/7 (MT626826) and *Pseudomonas* sp. strain COW5 (MT507079) were found to be the closest characterized relatives of the two aforementioned phylotypes (Table 1). Of them, the former has been isolated from skin ulcerations in cod (*Gadus morhua*) from the Baltic Sea (Pekala-Safinska et al., Unpublished data), while the latter was from the white cocoyam rhizosphere (Oni et al., Unpublished data). Also, the phylotypes *P. putida* (found in whole, gutted, and filleted fish) and *P. gessardii* (found in gutted fish) were found to be highly close relatives with two phylotypes, the *P. putida* strain SeaQual_P_B75 (MT626824) and the *P. gessardii* strain SeaQual_P_B791/1 (MT626825), that have been also isolated from skin ulcerations in cod from the Baltic Sea (Pekala-Safinska et al., Unpublished data). Additionally, a highly close relative of *P. fluorescens* strain UTB_111 (MT624739) which has been found in sediments from Byers Peninsula, Livingston Island in Antarctica (Rubiano-Labrador et al., Unpublished data), was also found to be abundant in gutted and filleted fresh fish in the present study. *Pseudomonas* spp. have been recognized as SSOs of various seafood from the Mediterranean Sea, such as gilthead sea bream [20,23,24], European sea bass [25,26,27,28], and Mediterranean boque [29], which were stored aerobically at low temperatures (0–10 °C). However, it has been observed that the domination of a *Pseudomonas* phylotype with higher growth rate and different metabolism than another phylotype (even of a high degree of homology of the 16S rRNA sequence), can sometimes affect sensory attributes and product shelf-life [18,19,30]. For this reason, the knowledge of phylotypes and their abundances or populations in fresh fish is of high priority for seafood microbiology specialists.

Shelf-life depends on the initial APC level, the composition of the initial microbiota of fish, microbial interactions, the exposure conditions of fish after catch (e.g., temperature and time), the storage conditions, etc. In fact, all these factors determine the domination of SSOs in fish and so the production of metabolites that further deteriorate fish quality (production of off-odors and -flavors) and lead to the rejection of the product [6]. Herein, *Pseudomonas* and H_2_S-producing bacteria were found to dominate over the other microorganisms tested during the storage, reaching spoilage population levels of 7–9 logs and causing the sensory rejection (fish determined as unfit) after days 15, 15 and 10 for the whole, gutted, and filleted fish, respectively. Indeed, such bacteria were also found to compose the cultivable sea bass microbiota using molecular approach. Although the initial microbiota of the whole and gutted fish was not similar in the present study, both products presented the same shelf-life (15 days) and the same potential spoilers (same phylotypes) such as *P. gessardii*, *P. putida*, *Pseudomonas* sp. (phylotype SBS-FS3), *S. baltica*, with *P. gessardii* dominating in both cases. However, all these bacteria were found at different abundances between the two products. This could be explained if we take into account that the gutted fish had been contaminated during primary processing, with some potential pathogens that probably exhibit different interaction behaviors than those present in the whole fish. Regarding sea bass fillets, the shelf-life (10 days) was much shorter than the other products (15 days) even though it was stored only 1–2 °C higher than them. Additionally, fish fillets presented significant differences on the microbiota composition both at the beginning and the end of shelf-life compared to the whole and gutted fish. To give clear answers on the aforementioned queries (e.g., on the much shorter shelf-life of the fish fillets in comparison to the whole and gutted fish), the spoilage potential of each isolated phylotype has to be elucidated in a future study.

The initial microbiota of sea bass also consisted of potential food-borne pathogens that are usually linked with contamination from the environment in handling and primary processing; for example, *K. oxytoca*, *S. fonticola*, *E. ludwigii* and *Rouxiella* sp. from the human or animal digestive tract, and *S. warneri* and *S. maltophilia* from human hands, nose, throat, etc. In our study, HRM gave similar melting curve profiles for *S. grimessi*, *S. warneri* and *E. ludwigii* in both gutted and filleted fish, but this was not observed for the whole fish (these phylotypes were not detected in the whole fish). Also, *Rouxiella* sp. was isolated from neither whole nor gutted fish. The presence of such bacteria in gutted and filleted fish compared to the whole fish indicates the effect of primary processing on the sea bass products microbiota since all fish originated from the same batch. Indeed, food handlers, food-contact surfaces, tools and equipment, washing water, other contaminated foods or ingredients, etc., have been found to contaminate foods with potential pathogens [31,32,33]. For example, Yap et al. [33] reported that the handling of sushi with bare hands significantly increases the population of total microorganisms and hygiene indicators such as *S. aureus* compared to handling by gloved hands. Also, proper hand washing in food handling and preparation minimizes the spread of pathogens [32]. Herein, the presence of potential foodborne pathogens that might be transferred from the environment to the gutted and filleted fish during handling, filleting, etc., reveals the urgent need for updated or new strategies to follow towards microbial safety and quality assurance in aquaculture and the fish processing industry. HRM could be used as a tool to reveal the contamination in fish in order for stakeholders to implement measures that improve the post-farm gate practices, e.g., by reinforcing Good Hygiene Practices. The rapid detection of contamination and the immediate solving of problems related with microbial quality or safety of fish will increase competitiveness of the products for the aquaculture sector and fish processing industry.

## 5. Conclusions

This study revealed the dominant phylotypes in the most common commercial sea bass products of the Hellenic aquaculture (whole, gutted, and filleted fish) and showed the effect of primary processing and storage on the products microbiota. Significant fish spoilers such as members of *Pseudomonas* (e.g., *P. gessardii*, *P. putida*, *P. fluorescens*, *P. fragi*) and *Shewanella* (e.g., *S. baltica*, and *S. putrefaciens*), as well as potential pathogens e.g., *A. hydrophila*, *K. oxytoca*, *S. fonticola*, *S. grimessi*, *E. ludwigii*, *Rouxiella* sp. and *S. warneri*, composed the cultivable microbiota of sea bass products at the beginning and/or the end of fish shelf-life. Moreover, differences were observed on the microbiota composition of the fresh and chill-stored gutted and filleted fish compared to the whole fish due to the contamination during handling and primary processing. Based on the findings, we highlight the urgent need for updated or new strategies to follow towards microbial safety and quality assurance in aquaculture/fish industry. This could be achieved through the development and establishment of new tools or toolkits e.g., intelligent microbial tracking systems that lead to the rapid detection of fish spoilers or hazards and the immediate implementation of the corrective actions in pre- and post-farm gate.

## Figures and Tables

**Figure 1 foods-10-00671-f001:**
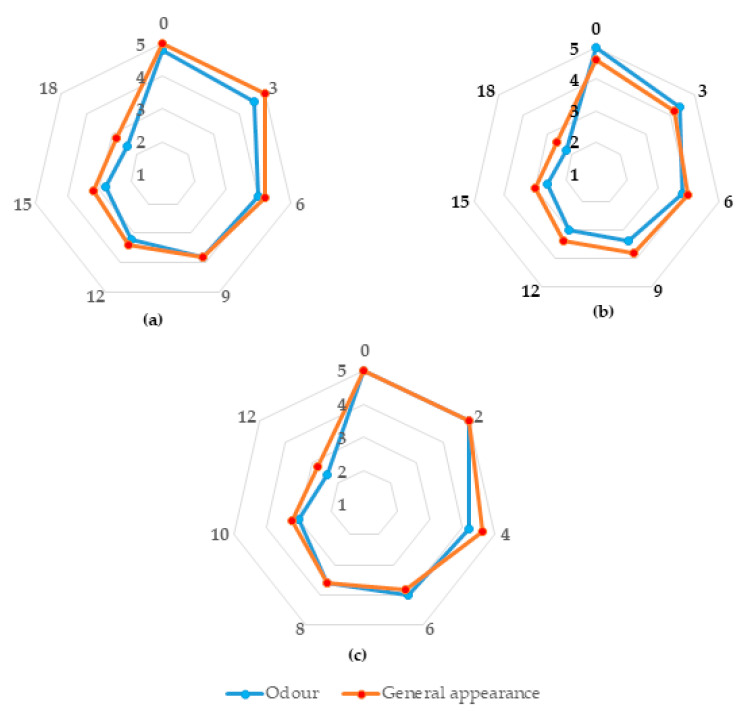
Sensory score of whole (**a**), gutted (**b**) and filleted (**c**) sea bass stored aerobically for 18, 18 and 12 days, respectively, at chilled temperatures. The numbers 5, 4, 3, 2 and 1 corresponds to a 5-point scale, with 5 indicating the highest freshness level and 1 the lowest (totally spoiled). A score of 3 was considered the score for minimum acceptability.

**Figure 2 foods-10-00671-f002:**
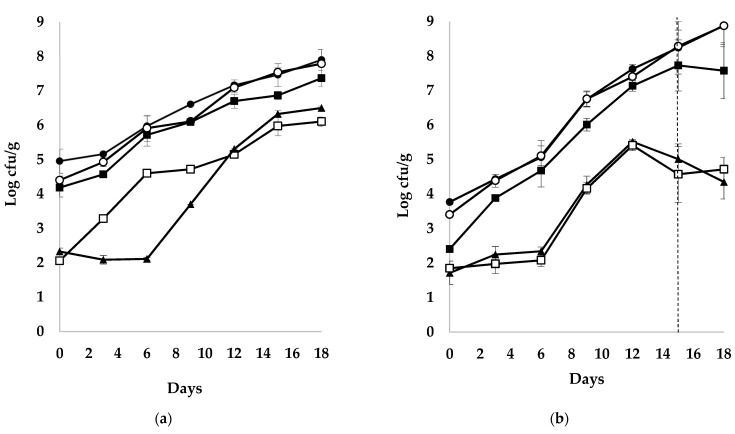
Microbiological changes of whole (**a**), gutted (**b**) and filleted (**c**) sea bass stored at chilled temperatures. APC (●), *Pseudomonas* spp. (○), H_2_S-producing bacteria (■), LAB (□) and Enterobacteriaceae (▲). Each data point and the error bars show the mean and ± st. dev. of 3 replicates. The vertical dashed lines indicate the end of fish shelf-life.

**Figure 3 foods-10-00671-f003:**
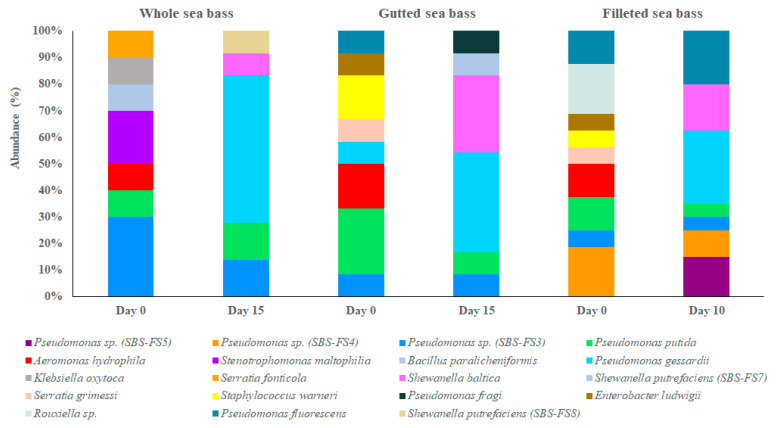
Microbiota of whole, gutted, and filleted chill-stored sea bass at the beginning and at the end of shelf-life, as determined by HRM and 16S rRNA gene sequencing analysis.

**Table 1 foods-10-00671-t001:** Melting peak temperature (mean ± st. dev of three replicates) and closest relatives of the 200 analyzed isolates.

Groups	Isolates	Peaks (°C)	Phylotypes	Closest Relatives	Identity (%)	GenBank Number *
1	51	85.3 ± 0.1	SBS-FS1	*Pseudomonas gessardii*	100	MT626825
2	23	85.25 ± 0.05	SBS-FS2	*Pseudomonas putida*	100	MT626824
3	21	85.3 ± 0.1	SBS-FS3	*Pseudomonas* sp.	99.75	MT626826
4	10	85.9 ± 0.1	SBS-FS4	*Pseudomonas* sp.	99.75	MT507079
5	6	85.5 ± 0.1	SBS-FS5	*P**seudomonas* sp.	100	MT585910
6	24	87.2 ± 0.1	SBS-FS6	*Shewanella baltica*	100	MT516290
7	4	86.8 ± 0.1	SBS-FS7	*Shewanella putrefaciens*	99.75	AB205575
8	3	86.8	SBS-FS8	*Shewanella putrefaciens*	99.75	MN865784
9	10	86.6 ± 0.1	SBS-FS9	*Aeromonas hydrophila*	100	MT416424
10	2	87.4	SBS-FS10	*Bacillus paralicheniformis*	99.74	MT645610
11	4	86.95 ± 0.05	SBS-FS11	*Serratia grimesii*	100	MN826574
12	2	86.8	SBS-FS12	*Serratia fonticola*	100	CP054160
13	2	87.9	SBS-FS13	*Klebsiella oxytoca*	100	MN967236
14	14	85.2 ± 0.1	SBS-FS14	*Pseudomonas fluorescens*	99.75	MT624739
15	4	86.4 ± 0.1	SBS-FS15	*Stenotrophomonas maltophilia*	100	CP040432
16	4	85.1 ± 0.1	SBS-FS16	*Pseudomonas fragi*	99.75	MT631986
17	6	85.7 ± 0.1	SBS-FS17	*Staphylococcus warneri*	100	MT642942
18	6	85.9 ± 0.1	SBS-FS18	*Rouxiella* sp.	100	MK590241
19	4	87.8 ± 0.1	SBS-FS19	*Enterobacter ludwigii*	99.75	MT507089

* Genbank numbers belong to the closest characterized relatives of the phylogroups identified in this study.

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
