# Peer review of "Primary Processing and Storage Affect the Dominant Microbiota of Fresh and Chill-Stored Sea Bass Products"

_foods, 2021, doi:10.3390/foods10030671_

Round 1
Reviewer 1 Report
The authors present an interesting study on 1) the cultivable microorganisms from whole, gutted and filleted sea bass products during chilled-storage and 2) the effect of primary processing on microbial communities in gutted and filleted sea bass products compared to the whole fish. Results of this work show that significant fish spoilers as well as potential pathogens were isolated from the sea bass products at the beginning and/or end of their shelf-life. Furthermore, differences in the microbiota of fresh and chilled-stored processed products were found compared to the whole fish. These results demonstrate the contamination of the gutted and filleted products during processing and highlight the need for appropriate strategies to ensure microbial safety and quality in aquaculture and fish processing industries.
The authors present work that will be of interest to readers from the aquaculture and fish processing fields. The study design is clearly explained and appropriate, the results are well-described and the authors' conclusions are supported by their results.
I note that the authors have addressed all of my comments from their first submission and I am happy with the way these comments were addressed. I only have one minor point for the authors to address:
Line 144-148: The authors should specify whether the default settings were used for the sequence alignments, and if not, what settings were applied. How do the authors define ≥98% homology? Do the authors mean that after alignment the sequences shared ≥98% sequence identity?
Author Response
Reviewer 1
The authors present an interesting study on 1) the cultivable microorganisms from whole, gutted and filleted sea bass products during chilled-storage and 2) the effect of primary processing on microbial communities in gutted and filleted sea bass products compared to the whole fish. Results of this work show that significant fish spoilers as well as potential pathogens were isolated from the sea bass products at the beginning and/or end of their shelf-life. Furthermore, differences in the microbiota of fresh and chilled-stored processed products were found compared to the whole fish. These results demonstrate the contamination of the gutted and filleted products during processing and highlight the need for appropriate strategies to ensure microbial safety and quality in aquaculture and fish processing industries.
The authors present work that will be of interest to readers from the aquaculture and fish processing fields. The study design is clearly explained and appropriate, the results are well-described and the authors' conclusions are supported by their results.
I note that the authors have addressed all of my comments from their first submission and I am happy with the way these comments were addressed. I only have one minor point for the authors to address:
Line 144-148: The authors should specify whether the default settings were used for the sequence alignments, and if not, what settings were applied. How do the authors define ≥98% homology? Do the authors mean that after alignment the sequences shared ≥98% sequence identity?
We used a cut-off level of 98% sequence similarity to GenBank entries to define the closest relatives of our sequences. It was corrected in the manuscript.
Default settings (0) were used for the sequence alignments due to the differences on the sequences size.
Reviewer 2 Report
The objective of the study is to study the cultivable microbiota of three sea bass products (whole, gutted and filleted fish from the same batch) during chilled storage and study the effect of primary processing on microbial community structure.
Major Concern:
The rationale behind the experimental design as well as statistical analysis of the development of the microbial populations are missing. Two different storage temperature is used (0 and 2 °C), will it affect the microbiological analysis and the analysis of the microbial community? How is the quality of the ice used for storage? Ice quality is essential as there are direct contact between ice and products. Is there a statistical difference in microbiological counts as a function of storage time between the groups? It seems like it is a statistical difference already at day 0. Please include statistical analysis.
The topic is of high interest for the scientific community and for the seafood industry as it addresses the effect of primary processing on microbial communities and conduct HRM analysis.
Minor concern:
Point 1: Line 62: I suggest including a short description of the High Resolution Melting (HRM) methodology
Point 2: Line 82: Storage time? Why is a different storage temperature chosen for the filets compared to the other products? How many fish in each group?
Point 3: Line 83-90: Evaluation of sensory rejection time should be described more precise. At which timepoints were the analysis done? Had the panellist any training in the methodology? Did they work independently? How the grading scale?
Point 4: Line 110, Revise section title
Point 5: line 111 “Colonies were obtained (30-50%) from TSA plates”. What is meant here? How many colonies were pick for each product and how was the selection done? Randomly picked?
Point 6: Line 120-125: In my opinion this information belongs to the Introduction part, not Material and Methods.
Point 7: Figure 1: The resolution is not adequate. The grading system must be explained in the Figure legends, as it is now the Figure give no meaning.
Point 8: Line 234: Figure 3: How many samples per group?
Point 9: Line 328-333: the authors states “HRM could be used as a tool to reveal the contamination in fish in order for stakeholders to implement measures that improve the post-farm gate practices, e.g. by reinforcing Good Hygiene Practices”. How rapid is the method? How easy and user friendly is the methodology for the industrial sector?
Point 10: I miss a link between the measured sensory spoilage, microbial counts, and the detected microbial communities in the discussion section.
Point 11: References: References section should be corrected according to “Guide for Authors” of “Foods”.
Author Response
Reviewer 2
The objective of the study is to study the cultivable microbiota of three sea bass products (whole, gutted and filleted fish from the same batch) during chilled storage and study the effect of primary processing on microbial community structure.
Major Concern:
The rationale behind the experimental design as well as statistical analysis of the development of the microbial populations are missing. Two different storage temperature is used (0 and 2 °C), will it affect the microbiological analysis and the analysis of the microbial community? How is the quality of the ice used for storage? Ice quality is essential as there are direct contact between ice and products. Is there a statistical difference in microbiological counts as a function of storage time between the groups? It seems like it is a statistical difference already at day 0. Please include statistical analysis.
Authors would like to thank the reviewer for the comment.
Statistical analysis was added in the manuscript (materials and methods, results).
The aim of this study was to study the effect of primary processing and storage. For this reason, the two different temperatures were used to simulate the real/commercial storage and distribution conditions which the fish industry uses (L 88-90).
The ice is produced by an ice-machine connected with potable chlorinated water.
The topic is of high interest for the scientific community and for the seafood industry as it addresses the effect of primary processing on microbial communities and conduct HRM analysis.
Minor concern:
Point 1: Line 62: I suggest including a short description of the High Resolution Melting (HRM) methodology
It was revised. Herein, we addressed the comment of the point 6 (L 63-69)
Point 2: Line 82: Storage time? Why is a different storage temperature chosen for the filets compared to the other products? How many fish in each group?
Storage time was added (L 87-88).
In Greece, the whole and gutted fish are stored in ice, while fish fillets are stored at 2oC. Thus, the two different temperatures were used to simulate the real storage conditions which fish producers and distributors apply. This was also clarified in the manuscript (L 88-90).
We used 3 whole, gutted, or filleted fish per time point (see L 119). This means that 3 fish x 7 time points per product (21 whole, 21 gutted and 21 fillets). It was added in the manuscript (L 119-120).
Point 3: Line 83-90: Evaluation of sensory rejection time should be described more precise. At which timepoints were the analysis done? Had the panellist any training in the methodology? Did they work independently? How the grading scale?
It was addressed (L 95-102)
Point 4: Line 110, Revise section title
It was revised (L 123)
Point 5: line 111 “Colonies were obtained (30-50%) from TSA plates”. What is meant here? How many colonies were pick for each product and how was the selection done? Randomly picked?
It was revised (L 124-127)
Point 6: Line 120-125: In my opinion this information belongs to the Introduction part, not Material and Methods.
It was addressed (L 63-69)
Point 7: Figure 1: The resolution is not adequate. The grading system must be explained in the Figure legends, as it is now the Figure give no meaning.
It was revised
Point 8: Line 234: Figure 3: How many samples per group?
The number of samples are shown in L 119-120 (3 fish per product and time point, meaning 3 whole, 3 gutted and 3 filleted fish at day 0, and 3 whole, 3 gutted and 3 filleted fish at the end of shelf-life).
Point 9: Line 328-333: the authors states “HRM could be used as a tool to reveal the contamination in fish in order for stakeholders to implement measures that improve the post-farm gate practices, e.g. by reinforcing Good Hygiene Practices”. How rapid is the method? How easy and user friendly is the methodology for the industrial sector?
This method can give results in approx. 2 hours. We can differentiate bacteria based on the shape of HRM curves and the identification can be performed based on the shape of the reference strains (Syropoulou et al 2020) or sequencing. Fish industry will use a toolkit that we prepare, containing analytical guidelines for ‘automate’ HRM analysis and step-by-step guidelines for the implementation. Consulting services are also available to help stakeholders towards problems solving related with microbial quality and safety of seafood.
Point 10: I miss a link between the measured sensory spoilage, microbial counts, and the detected microbial communities in the discussion section.
Some lines were added in the manuscript (L308-312).
Point 11: References: References section should be corrected according to “Guide for Authors” of “Foods”.
It was revised